# Communicating Terroir through Wine Label Toponymy Greek Wineries Practice

Theodosios Tsiakis [1] , Eleni Anagnostou [1], Giuseppe Granata [2,*] and Vasiliki Manakou [3]

1   Department of Organisation Management, Marketing and Tourism, International Hellenic University, 57400 Thessaloniki, Greece
2   Department of Economics, University Mercatorum of Rome, 00186 Rome, Italy
3   School of Civil Engineering, Faculty of Engineering, Aristotle University of Thessaloniki, 54124 Thessaloniki, Greece
*   Correspondence: giuseppe.granata@unimercatorum.it

**Abstract:** For the majority of consumers, the label is the primary motivation for wine purchases. It appears from the literature that consumer behavior is influenced by the variety of information on the label, which captures with simplicity and clarity, the key information that the potential buyer needs to know. History, place, variety, name, and figure, are some of the basic elements that form the wine label and have the potential to profoundly affect consumer engagement with bottled wine. What do Greek Wineries wish to communicate through their wine labels? Historical concepts, identity, or quality? Could soft power and place branding be suitable factors to help transfer this message and achieve wishful feedback to the consumer's awareness? A case study is presented, according to which Greek producers select three fundamental cues: toponymy, name description, justification of the name choice and language. Driven by this case study, the paper intends to open a discussion about the implementation of theories such as soft power and sense of place by wine industries on a global level in combination with the importance of the toponymy, not only on the labeling but also on other communicational aspects.

**Keywords:** vineyards; place names; wine label; toponym; place name; place branding; soft power





## 1. Introduction

The current antagonistic economic environment creates intense competition among wineries to meet customer demands. Synchronously, we observe a tremendous advancement in market mechanisms to reach and support customer needs. Companies and organizations innovate by co-creating value with customers and utilizing knowledge about users (user-driven innovation) along with knowledge supply and collaboration networks. Communication and marketing strategy are the primary mechanisms to better compete in the complex market environment. Wine communication forms the strategy of transferring and describing everything that tasting struggles to divulge. However, most of all it communicates the place that was the nascence of wine (called the "terroir"). Communicating wine is always a constant challenge, especially in terms of the composition and organization of that message that will be able to transfer in the most appropriate way the particular terroir of each vineyard. This is because wine is a special product. Understanding terroir is crucial because it conveys the context of touristic attraction for Greek wineries. How can we bridge this gap? The solution comes from toponymy which gives us a vast and unexploded field of research. The toponym can enable the potential consumer of the wine to understand the uniqueness of the vineyard, its identity and much more the philosophy of the winemaker.

A winery is a multidimensional creation cell. In addition to its creative, social and cultural dimensions, it must also function as an organized economic unit with strategic reference and management. The quality of the wine is the key to success, but quality equals terroir, which is the most important factor that determines its value. Toponomy could

be used as a 'communicative mirror' of what a specific destination can offer to tourists. The place name can be transformed into a communication mechanism that diffuses the wine terroir and the specific character of an area as a brand name destination. In this logic, a wine label should automatically reflect the special aroma, the climatic conditions, the peculiarity of the soil and the indigenous grape varieties, of the brand-name vineyard, which can become a successful brand destination. The communication strategy must be formulated in such a way that the vine's name could attract tourists who wish to consume a holistic experience: special features, facilities, activities, entertainment and socializing.

The wine label is a reflection of the winemaker's cognizance of the wine (it reveals the way in which the winemaker discourse with the market and his customers). The name and the logo consist the most important features in the image of wine, but there are many other things, fancy words such as Unfiltered, Old Vines, Single Vineyard, Wild Ferment, Sur Lie that try to indicate the distinctiveness (which is often true) and that naturally justify an added value. The names of the label generally result either from grape variety (the grape used to make the wine) or from the region in which the wine was made. A third pop-up is «terroir» deriving directly, when appropriate, from the site name of the vineyard plot. Wines in the Old World, generally receive their names from the region from which the wine was made, Bordeaux region of France as an example (regional style). In the New World side, the majority of the winemakers give their names according to the sole or principal grape varietal (varietal style). The reason wineries from the Old World name their wine after regions is because they want to emphasize the sense of the place called "terroir" as it is believed that the same grape can have different characteristics based on climate, soil and terroir. Further differentiation can be made between literal and figurative naming strategies. The literal strategy is notable mainly for its use of toponyms (or place name) [1]. Wine labels and their associated information has shown that they represent useful information that influences consumer choice [2], where front labels were found to have more power than back labels [3], and an inclination to young consumers [4].

The present work unfolds, both through the theoretical basis and through research, the communicative approach of the terroir and its promotion through the visual communication of the wine label with the toponym as the core of interest. It is the specificity of the toponym that creates the research interest in how wineries can be strengthened financially, communicatively and commercially by using the toponym as a means of conveying the key perceptions of winemakers about their wines. Each label and each name is a combination of signs, conveying important information about the history, variety, philosophy, identity and specificity of each bottle of wine, each producer and each wine region. On this axis, the paper tries to synthesize different approaches to toponymy and examine how the vineyard names can become commodified symbols of terroir, in relation to the wineries' communication strategies. Moreover, it attempts to modify the way that Greek Wineries could become attractive brand-named destinations as toponymy (or place name) operates catalytically in expressing terroir information.

More specifically, the paper analyzes the specific characteristics of Greek wines and the reasons why language and communication are necessary for relating to a wine label. Then it presents the relationship which is developed between the place name and the terroir and what this means for wine communication. In the following section, the theories of sense of place and soft power are proposed for the application of communication in wine businesses. The last section presents our case study about the application of toponymy by Greek wineries and summarizes the main conclusions of our work.

## 2. Specific Characteristics of Greek Wine

The geomorphological and structural elements of the market are instrumental in the identification of characteristics of Greek wine. Greece is a small country, with complicated topography, two elements that demonstrate the difficulties for Greek wine to be antagonistic both in volume and price. Greece is a micro-producer on the scale of vineyards as the extent of Greece's vineyards is about 106 thousand ha (to understand this, Bordeaux, is

twice that size and it's only one wine region of France) producing 2.2 million hl wine [5]. Greek wineries can be divided into four main categories: large wineries with a production capacity greater than 100,000 hl per year, medium with a production capacity of 30,000 up to 100,000 hl per year, in small, usually family wineries with limited production capacity (less than 30,000 tons) and cooperatives that produce and distribute wine, mainly locally.

A key issue of Greek wines is mispronunciation (hard-to-pronounce names) of Greek varieties such as:

- Xinomavro [ksee-NOH-mah-vroh]
- Agiorghitiko (eye-yor-YEE-tee-koh)
- Assyrtiko (ahs-SEER-tee-koh)

In Greek, for example, the letter β is pronounced: "veeta", say "VEE-tah". A 'B' makes a 'V' sound in the Greek alphabet, the letter 'β' is pronounced as English speakers pronounce a 'V'. A "BEE" phoneme in Greek "μπ", is used as in "μπαλσάμικο" ("balsamiko", balsamic). However, in science, people use "beta", η "eeta", ι "yiota", μ "mee", ν "nee", π "pee", τ "taf", χ "hee", ψ "psee". The difficulties in pronouncing the γ sound and such, and the translation of the "ee" sound into "i" is what people do not understand. Moreover, the capital letters B, H, P, X and Y look similar to the alphabet used in English, but in Modern Greek they are pronounced differently. The transliteration of place names from Greek to English as a lingua franca (ELF) is an issue of great importance.

There is no doubt that the correct spelling in Modern Greek and its corresponding transliteration into Latin characters is fundamental. The attribution of a place name must be in such a way that it can lead us "inverted" from the Greek rendering to the true form of the name as closely as possible. It should lead, similarly, to the orthographic expression (and where possible to the pronunciation) of the name. The adoption of ISO 843/ELOT 743 is proper for Greek wine. The Greek state also used it in the transliteration of addresses {i.e., Greek: "Μεγάλου Αλεξάνδρου" address if formed into "Megalou Alexadrou" and not "Alexander the Great"}. The interdependence of and interaction of the various components with one another indicate the complexity.

## 3. The Importance of Language and Communication

Communicating wine is oriented toward end consumers, professionals and intermediaries (including journalists, restaurateurs, wineshops and wine-bars) with different marketing tools in different situations [6]. The front label of a wine bottle is a purposeful approach specifically as the first point of contact with consumers dispensing implications about marketing communication [7]. It consists of the primary information source and factors for wine-buying decisions, overlooking the issue of design [8]. Even the back labels of wine have a significant effect on consumers' expected liking, informed liking, wine-evoked emotions and willingness to pay [2]. The label generally is a prime marketing tool, consisting of a key source of information for the purchaser [9]. Moreover, from all the information provided on the label, the name (of the product) is the principal means of summarizing to the consumer precisely what a product stands for [10].

Front labels present essential information for the consumer, including the wine and winery name, and the back label sensory characteristics, winery history information, and food pairings. For medium- and higher-priced wines, back label information should include winery history information (information indicating terroir) [11]. However, the way consumers evaluate label information credibility or trustworthiness still remains limited [12]. Affirmative requirements in wine labels intend to provide consumers with the information they need to make informed choices [13]. König and Lick [14] showed that wine labels are positively correlated with price-sensitive customers and also that wine producers use specific oriented names as brand names of their wines in order to emphasize the origin of wines. An example of a confusing labeling strategy is the case of organic wines where organic wines can bear two labels, but the meaning of the different terminology is not further explained on the label [15].

Label is a simple term meaning "a display of written, printed, or graphic matter upon the immediate container of any article"; labeling is a broader term, containing labels and other visual data on a product [16,17]. Labeling along with packaging are the cues that consumers use when they choose wines [18]. A wine label must include the brand name, class of wine (type, vintage date, the appellation of origin, etc.), place bottled or packed stated as, "bottled by" or "packed by" or "imported by" for imported wine, the net contents, declaration of sulfites, alcohol content declared as "Alcohol _% by Vol. [12]. Visual attributes of the packaging such as the colors are often used to evaluate their impact on the intent of purchase or perceived quality [19]. But labels (as information mean), are considered to be among the most important cues consumers use in the wine choice decision [18]. Considering that wine is a hedonic product, the use of names that notate high-quality features may be important [20]. So, the notion of a uniform label is less profitable for high-quality producers who seek to establish a high degree of diversification in a region [21]. Distinctive labels can only be carried on wines that originate from a highly specified area [22]. A "good" brand name according to Medway and Warnaby [23] should be: (i) simple (ii) distinctive; (iii) memorable; (iv) meaningful (v) evocative, (vi) protectable and (vii) transferable.

## 4. Toponymy Linking with Terroir

Vines are shaped by three environmental factors: climate, topography and soil [24]. The element of topography emerges as the bridge between soil and climate in the terroir equation (hills, valleys, and orientation to the sun have significant viticulture effects resulting in the relative quality of wine) [25]. People use toponymic names in order to express these topography characteristics so the practice to use it in a wine label is obviously to demonstrate the connection of the land (terroir) with the product (wine). However, there is a necessity to refine positioning through differentiation criteria such as topography, as terroir alone soon will lose its ability to add value to a wine [26]. The land with its elements is the one that influences the flavors of food and beverages, but ultimately the cultural domain creates the goût du terroir [27]. The element of "terroir" is the one that produces added value as it offers value propositions to producers because opponents cannot reproduce the terroir and consumers that cannot have this "flavorsome" from anywhere else [28]. It shows the link between taste and place, for example, Burgundy wines have different taste profiles than wines from Bordeaux [27]. The problem with using the concept of terroir as a unique selling point is that the consumer stared in perplexity at the notion [29]. The terroir is used by producers as a marketing mechanism in order to manifest the distinct wine as different from all others due to its place of origin [30].

The International Organisation of Vine and Wine [31] defines vitivinicultural "terroir" as "a concept which refers to an area in which collective knowledge of the interactions between the identifiable physical and biological environment and applied vitivinicultural practices develop, providing distinctive characteristics for the products originating from this area". There has been a long way for the French word terroir to refer to "the complex interaction between all of the physical aspects of geology, soils, climate, geomorphology and vegetation that combine to create a particular 'place' where grapes are grown" [32] as of two decades ago viticultural terroir was used infrequently to define a region related to a particular area with an exceptional quality of grapes and wines [33]. Terroir is characterized as the intersection of four components of territory, plant growth, advertising, and identity [34]. In terms of biology, terroir reflects differences in fruit composition caused by growing the vine in a different environment [35]. It is crucial to create fundamental bonds of wine place with identity in order for the consumer to develop a deeper appreciation of the wine [36]. Moreover, the notion of landscape influences the preference of consumers for wine and is useful in market segmentation [7].

The notion of place names as a brand name interrelates toponymy and branding [23]. Topographic, geologic and soil features and human activities are usually used for a place name. In Greece, a plethora of toponyms is related to the specific unique characteristics

of these features. Toponymy derives from the Greek words' topos ('place') and onoma ('name') and is considered the systematic study of the origin and history of place names [37]. Britannica [38] defines toponymy as: "the taxonomic study of place-names, based on etymological, historical, and geographical information" divided into two broad categories:

1.   habitation names—a locality that is peopled or inhabited
2.   feature names—natural or physical features of the landscape

The value of toponymy may be archaeological, historic linguistic and folklore. A place name may owe its name to:

- a predominant feature
- its agricultural use,
- a historical or mythical person or event
- technical projects
- any other position or feature
- the names of gods, saints or heroes,
- names of animals and plants (zonyms—phytonyms), e.g., Melissia, Daphne, etc.
- popular perceptions and beliefs

The difficulty in translating a toponym, lies in its nature as a sign, the sign we fight for, the sign that has layers of meanings we do not want to be forgotten [39].

According to Tent [40], there are two basic ways to perform toponymic research, an etymological analysis and an original analysis of toponyms. Onward research of Tent [41] resulted in an additional coding: Indigenous—containing at least one indigenous element, Introduced—no indigenous generic or specific elements. Many attempts have been made for place names to be classified in a systematic manner and Stewart [42] was the first researcher that set the theory that all placenames arise from a single motivation, the desire to distinguish and separate a particular place from places in general, defined in ten toponym types [43]:

1.   Descriptive names
2.   Associative names
3.   Incident-names
4.   Possessive names
5.   Commemorative names
6.   Condemnatory names
7.   Folk-etymologies
8.   Manufactured names
9.   Mistake-names
10.  Shift-names

Stewart's toponymic typology has been criticized for being too broad and inconsistent and other scholars have attempted to improve it [44,45]. However, in our case, Stewart's classification serves as well as it interrelates well with wine toponymy names which are based on data from databases and allow a quantitative analysis of the frequency of occurrence of place names.

## 5. Proposing the Theories of Sense of Place and Soft Power for the Wine Industry

The communicative reflections of toponomy in terms of capturing the terroir of Greek wineries could be relayed on two basic theoretical backgrounds that give us many possibilities: soft power and place branding. The approach of soft power in the field of communication strategy is very interesting, because of its important factors that can have a profound implementation on the use of toponymy. Chitty [46] claims that soft power is nice as it relies on cultural and public diplomacy. Communication, collaboration, values of clarification and decision-making are crucial to the improvement of soft power, a theory that has to embrace in order to co-opt and obtain the ability to shape preferences, interact and influence our purpose. Chitty [47] also points out the importance of soft power as well as the role of civic virtue in soft power aspirations and focuses on three categories of

mechanisms: mobility, media, and cultural industries. Chitty's work declares that social media gives the ability to ordinary people produce and share content that could make their voices be heard, not to mention that the categorization of the active soft power resources offers food for thought regarding public diplomacy. Furthermore, it concludes that certain types of communication generate soft power: dialogue is more attractive to ordinary people than strategic communication, which could be more suited for allies during a crisis.

Zaharna [48] declares that those who master message exchange will command communication power and that the strategic power of networks has become the new model of global persuasion. Zaharna also analyzes a holistic system based on communications all over the world, in which soft power is being created by network communication, which absorbs and integrates cultural diversity and gives the opportunity to the mass media to remain the dynamic players in the global system. Chitty [49] predicts that the next great technologically driven transformation of international communications is likely to take place after 2025 when humankind will enter new dimensions of space exploration. According to Nye [50], information once reserved for the government is now available for mass consumption and the internet has literally obtained new power. In the field of liberal values, virtue is related to the creation of messages, cultural events, objects, and programs and these are important elements of the key to soft power [46].

The place branding approach is based on the general idea that people create strong and emotional bonds with places. This is what Kavaratzis and Hatch [51] mentioned exactly as the answer to what place branding could mean in the field of branding industry and popularity. Their work also emphasizes the special importance of impression in relation to relevance, content and honesty and their research could be implemented in the Greek wineries case. Kavaratzis and Hatch [51] categorized place branding communication into four levels: (1) effective place branding expresses the place's cultural understandings, (2) leaves impressions on others, (3) changes identity by implanting new meanings and symbols into the culture and (4) mirror their impressions and expectations. According to their point of view, there are interactions between the physical and emotional dimension that explains the proper way of place branding.

According to Baker [52], to start with the part of fame and impression that leaves a place, stakeholders and leaders, are first of all the ones who have to understand the direct connection that exists between the image of a place and fame, as well as the attractiveness as a destination. There is a gap and heterogeneity between the internal identity of a place and its external reputation to the public. This is because each place identifies itself as a specific destination, but in practice does not meet the high standards it is supposed to have. This is a fact that should be taken seriously by stakeholders.

Aitken and Campelo [53] consider that understanding the connection between people and place is crucial for the development of a place brand because individual ideas are shaped when there is a shared perception in the community. Aitken and Campelo's work is related to the idea that a place can be developed as a brand when the sense of ownership has a social construction and conveys practices that are characterized by identity and culture. Aitken, R. and Campelo presented a theoretical model which identifies the foundational features of a place brand: four basic elements are very much of importance especially when they interact. Rights, roles, relationships and responsibilities are the factors of their model that creates brand awareness, brand authenticity and brand sustainability. It is also important to acknowledge that their contribution is for the better understanding of what place branding is in terms of co-creation, which can make the communities develop a brand identity that can lead to brand essence and commitment from stakeholders.

Grenni and Horlings [54], emphasize that branding and planning can help each other as long as place branding will be developed endogenously by a multi-stakeholder process and adopts the term 'together', which will support sustainable perspectives for the future. As Govers [55] puts it, places are complex. According to Govers a common mistake that is made is when places are mistakenly treated as products. It is quite the opposite that should be conducted, as he thinks that places offer environments that offer products that can

travel to the international markets, exactly as with tourism product market combinations or cultural offerings. Govers argues that places are spaces where people have a full life, enjoy their free time and involve themselves in social and cultural activities. Govers' assumptions are that a simple logo or a slogan won't make the difference, because place branding is much more than that, it is the deep consideration that it should be built on a sense of belonging and shared purpose and hence generate the kind of engagement that is desired and impossible to imitate elsewhere.

Researching the literature on communication and place branding, there is no doubt that drastic measures and changes in the basic way of thinking must be taken. The modern approaches of soft power and place branding lead us to redefine the way of communication on both levels of international and communication strategies. This could be a very promising intersection with the exploitation of toponymy as attractive place branding. Chitty [56] summarizes that state actors, international organizations, and multinational corporations should work towards a free flow of communication, which takes into account the pluralist nature of global society and addresses that pluralist nature. It is very important that he underlines that at the level of the individual, empowerment should increase one's ability to balance.

Bérard [57] suggests that "terroir must be viewed in a global context" and underlines the importance of understanding the link between product and place. Despite the fact that terroir as a principal stands opposite the local identity of a territory, nowadays it must be reconsidered in a new dimension with global expectations. This approach has social extensions and is linked to the study of the social construction of a sense of place [58], as place reflects much more than a structured frame. A place becomes part of experiences with family, friends and relationships in general. Extending this interpretation, what differentiates the concept of sense of place giving it some specificity, is not the preservation of historicity, but all the social relations of which it consists and to which it refers [59]. The timeless controversies regarding the creation of the sense of place refer on one hand to the perception of the physical environment and on the other hand to the personal and social perceptions of the place. Shamai and Ilatov [60] have tried to answer this question and demonstrated that the sense of place could be measured in an empirical way. They classified different models for measuring the sense of place theory, which is related to cultural and heritage backgrounds. Their bipolar method attributes to the positive scaling and negative scaling of the sense of place, declaring that: "places can be sensed differently by different groups". The implementation of the sense of place theory could be a different communicational approach for the wine industry globally, taking also into consideration the importance of toponymy. Useful insights indicate that branding has to deal with the comprehension of consumers' perception of the place's origin and brand names as well. Another structural element of the sense of place theory is the historical aspect and specifically the brand heritage. Places have a brand heritage that acts as a competitive advantage adding value to their products. In this sense wine industry and places have the same relation, but it is true that place heritage, corporate heritage and brand heritage may often interconnect but still remain different elements [61].

Based on the theoretical framework we consider that there are many possibilities for the combination of soft power and a sense of place. As Shamai and Ilatov [60] state: "places can be sensed differently by different groups". It is this element of a specific character that stems from the linking with a place and makes easily the understanding of origin from provenance [57,59]. Through the communication processes of persuasion, cultural diplomacy and brand heritage (soft power aspects) alongside the cultivation of the sense of place wine industries could evolve in a multi-layered way and improve their ability to contribute, shape and influence consumer preferences that reflect their culture, heritage, values and perception of what a place means to them.

### 6. Case Study: Application of Toponymy by Greek Wineries

Deciphering the toponymy of Greek wine labels is exciting, and adventurous but synchronously strenuous and daunting due to what we are about to deal with finding information sources and the ease of approaching and having feedback from a winery. Extended research was conducted to collect data mainly from wineries' websites and supplementarily through wine infomediaries, electronic wine shops, social media (Facebook, Twitter, etc.), national wine associations, newspapers and wine magazines. When the results of provided information regarding toponymic names (history and etymological meaning) were limited we turned to the solution of personal communication (by mail and telephone) with the wine producer or winemaker of the winery, so that they could assist us with the needed details and confront this information limitation. The typical sample was limited indicating that wine producers are in the beginning, starting to realize the notion of toponymy. In the following section, we present the winery along the label that has been given the toponymic name.

In Greece, there are about 700 active wine producers. (Wine producers with more than one winery are registered once, where their headquarters are settled). Active refers to those producers who already produce bottled wine. This number includes wine producers who have vineyards but they might not yet own a complete winery and are supported by other wineries. The sample consists of all wine labels presented electronically, including each type (example color) and each label is considered to be unique although it may share the same name from the moment one element is different (for example, white, rose and red are three separate wine labels even if they have the same name). The sample of wine labels we examined was numbered to 3487. The search has been conducted between May 2021 and June 2021.

In order to be included in this review, wine labels had to meet the following inclusion criteria:

- We evaluated whether the wine label name was defined as vineyard toponymy or not.
- The wine label name that did not explicitly state the reference to vineyard toponymy, were excluded. The same stood for the reason that the provided information was limited and there was no communication link to establish knowledge and evidence.
- Wine is not a single block, so it automatically loses its status, as it contains grapes from other vineyards or the vineyard occupies an area beyond the specified place name.
- By the time the research was conducted, the information source was either limited or the accomplishment of communication with the winery failed to succeed.
- The place name is complementary and not dominant; the unique identification of the name. That means the label name is a sum of three and more words.

We could have excluded the cases where toponymy is translated in English from Greek and thus the content and the reference substance of the place name are altered, but from the moment it is still an identifiable element of the label we decided to include it in the sample. These limitations resulted from 3487 examined labels only 49 fulfill the criteria. Table 1 represents wineries that incorporate toponymy as a wine label, the category of toponymy and a descriptive analysis according to the information found (as indicated above).

**Table 1.** Analysis of Greek Toponymy labels.

| ID | Winery | Toponymy Label | Toponymy Category | Description Analysis |
|----|--------|----------------|-------------------|----------------------|
| 1. | AcroTerra Wines | Skafida | Descriptive | Skafida—(>Eng. "cistern"). In Greek ("σκαφίδα") is a rectangular wooden or tin washtub where animals drink water or eat. Skafida is settled near the place where animals rest. The vineyard was that place for the animal to rest and feed. |
| 2. | Afianes Winery | Litany | Occurrent | Litany—Comes from the Greek word ("λιτανεία"). The Greek island Ikaria is famous for its Panigiria (fests), which include Saint's religious image litany around the village's streets. These special events are celebrated on the occasion of the name day of the church's Saint. |

**Table 1.** *Cont.*

| ID | Winery | Toponymy Label | Toponymy Category | Description Analysis |
|---|---|---|---|---|
| 3. | Alpha Estate | Barba Yannis | Eponymous | Barba Yannis—Barba (>Eng. "mister") + Yannis (>Eng. "John") = Barba Yannis (mister John). Mister (Barba) has a bilateral meaning. Comes from the Latin word ("barba") meaning the beard, a symbol of maturity, and therefore, respect. In Greek ("barba") means also uncle. This wine's name honors the last owner Barba Yannis, who planted the single block vineyard in 1919 and sold it in 1994 to the current owners. |
| 4. | Alpha Estate | Tramonto | Occurrent | Tramonto—(>Eng. "sunset"). The name refers to the vineyard's specific location, near the sun. The word Tramonto stems from Italian. The Italian influence as well as the impact of the so-called Latin languages is undoubtedly decisive. |
| 5. | Alpha Estate | Kaliva | Descriptive | Kaliva—(>Eng. "hut"). The term results from the ancient Greek (καλύβη/ kalúbē). In the area used to be an old hut. |
| 6. | Alpha Estate | 'Strofi' | Associative | Strofi—(>Eng. "turn"). The name indicates the characteristic shape of the vineyard which is a swivel. It could also indicate a metaphoric meaning as ("στροφή") in Greek means change. |
| 7. | Alpha Estate | Vrachos | Descriptive | Vrachos—(>Eng. "rock"). The initial meaning was the shallow waters of the sea combined with the steep rocky shores. A lake situates the vineyard! |
| 8. | Alpha Estate | Turtles | Associative | Turtles. This name was given due to the fact that the vineyard consists of a nesting site for the local turtle population. The name attributed translated. |
| 9. | Alpha Estate | Hedgehog | Associative | Hedgehog. The estate cares for hedgehogs as they are considered to be protected species. The vineyard was a nesting place for hedgehogs in ancient years. The name attributed translated also! |
| 10. | Artemis Karamolegos Winery | Papas | Eponymous | Papas—(>Eng. "clergyman"). Apparently was the owner of this resource and gave his name to the vineyard. |
| 11. | Artemis Karamolegos Winery | Louroi—Platia | Eponymous | Louroi-Platia—Societas Iesu settled in Santorini in about 1642 and used the word Louro (from Lura, the color of leather, the dark blonde; golden) for the characteristic color of Santorini land. Platia {πλατιά}—stands for widely, as the area is giving the essence of space. |
| 12. | Dalamaras Winery | Paliokalias | Descriptive | Paliokalias—palio (>Eng "old") + kale (Turkish word) >Eng ("castle—watchtower") = Paliokalias (the old watchtower). The term "καλέ" has been retained in the name of many fortifications in our country. The area was used by the Turkish as a watchtower. |
| 13. | Douloufakis winery | Aspros Lagos | Associative | Aspros Lagos—Aspros (>Eng "white") Lagos (>Eng "hare") = Aspros Lagos (white hare) is a toponym of vineyards named after the hares and "Asperoula". Asperoula is a wildflower that grows in the area and is also an endemic plant of Crete (Asperoula rigida M.). During spring the dry stones around vines are filled with Asperoula's white flowers. Their stems are hares' favorite food, and therefore, prefer the area for building their nests. |

**Table 1.** *Cont.*

| ID | Winery | Toponymy Label | Toponymy Category | Description Analysis |
|---|---|---|---|---|
| 14. | Hatzidakis Winery | Mylos | Associative | Mylos—(>Eng. "mill"). Owes its name to the homonymous vineyard located at an altitude of 220 m. In the village of Pyrgos in Kallisti (it is the highest village of Santorini) where traditional mills ("μύλοι") mylos (>Eng. windmills) of the 19th century have been built. They were settled on the ridge, and made of volcanic stones, water, soil and lime. Their height reaches about six meters. |
| 15. | Hatzidakis Winery | Louros | Descriptive | Societas Iesu settled in Santorini in about 1642 and used the word Louro (from Portuguese Lura, the color of leather, the dark blonde; golden) for the characteristic color of Santorini land. |
| 16. | Hatzimichalis Domaine | Drisbay | Descriptive | Drisbay—dris (>Eng. "oak") + bay (>Eng. ("master") = Drisbay ("the oak master"). The term means the master of the area. "Bay" is a Turkish linguistic influence. |
| 17. | Hatzimichalis Domaine | Yataki | Descriptive | Yataki—(>Eng. "bed"). "Yatak" is a linguistic Turkish influence that means a place to sleep or to live. |
| 18. | Hatzimichalis Domaine | Yerakofolia | Descriptive | Yerakofolia—yeraki (>Eng "hawk") + folia (>Eng "nest") = Yerakofolia (Hawk's nest). A place where hawks build their nests. |
| 19. | Hatzimichalis Domaine | Alepotrypa | Descriptive | Alepotrypa—alepou (>Eng "fox" + trypa (>Eng "hole" = Alepotrypa (Fox's hole). A characteristic sign of the plot. |
| 20. | Hatzimichalis Domaine | Kryovrisi | Associative | Kryovrisi—krya (>Eng "cold") + vrisi (>Eng "fount" = Kryovrisi (cold fount). In Greece and mainly in mountainous Greek villages there is a variety of such founts with abundant drinking water. |
| 21. | Hatzimichalis Domaine | Houlevena | Associative | Houlevena—xalevo (>Eng. "seek") + veno (>Eng. "go ahead") = houlevena (go ahead and seek). |
| 22. | Karimalis Winery | Kalabele | Descriptive | Kalabele—kalo (>Eng "good") + ampeli (>Eng "vineyard" = Kalabele (the good vineyard). This is a landmark name for an ancestral field of the Karimalis family, which is considered to maintain the ideal conditions for a vineyard. |
| 23. | Kir-Yanni Estate | Samaropetra | Descriptive | Samaropetra—samari (>Eng "saddle") + petra (> Eng "rock") = Samaropetra (the rocked saddle). That vineyard is placed on the top of a solid rocked layer, which refers to a packsaddle. |
| 24. | Kir-Yanni Estate | Ramnista | Descriptive | Ramnista—ramnos (>Eng. "white buckthorn") + istas (greek grammar suffix indicates the subject). In local dialect, ramnista means "slope that starts uphill". Ramnos (white buckthorn) is a deciduous shrub known as hippophae. The area used to be full of buckthorns, before its conversion into a vineyard. |
| 25. | Kir-Yanni Estate | Droumo | Descriptive | Droumo—from Greek "δρόμος"—dromos (>Eng. "road"). The vineyard is right on the main road of the area. |
| 26. | Kir-Yanni Estate | Palpo | Descriptive | Palpo—from Greek «πάλλω» "pallo" (>Eng. "pulsate") + po (linguistic suffix indicates the word as a noun) = pulsate. The vineyard is close to the train tracks and gets pulsated when trains cross the area. |

**Table 1.** *Cont.*

| ID | Winery | Toponymy Label | Toponymy Category | Description Analysis |
|---|---|---|---|---|
| 27. | Kokkinos Winery | Paliokalias | Descriptive | Paliokalias—palio (>Eng "old") + kale (Turkish word) >Eng "castle—watchtower") = Paliokalias (the old watchtower). Turkish word meaning castle- watchtower. The term " "καλέ" has been retained in the name of many fortifications in our country. The area used to be by the Turkish as a watchtower. |
| 28. | Lyrarakis winery | Armi | Associative | Armi—"αρμί" is a word in the Cretan dialect that defines mountainside top. The vineyard "Armi" is indeed a plot that lies at an altitude of 500 m. |
| 29. | Lyrarakis winery | Plakoura | Associative | Plakoura—plaka (>Eng. "flagstone") + oura (linguistic suffix indicates a superlative form). = plakoura (a wide and large stone). |
| 30. | Lyrarakis winery | Gerodeti | Associative | Gerodeti—Gero (>Eng. "old") + detis (>Eng. "stone") = Gerodetis (the old stone). In the Cretan dialect it is an old stonewall that joins or separates two fields. |
| 31. | Lyrarakis winery | Pirovolikes | Associative | Pirovolikes—from pyrovolo (>Eng. "firearm") + ikes ("linguistic suffix) = pirovolikes, referred by war with firearm ("pyrovolo") indicate that the location is a very important protection zone whilst it constitutes a very good control base for the whole area. |
| 32. | Lyrarakis winery | Voila | Descriptive | Voila—(>Eng. "landholder"). The word's origin comes from the Byzantine name "Voilas" as it was called by the Byzantines the landholder. |
| 33. | Lyrarakis winery | Aggelis | Eponymous | Aggelis Aggelos (>Eng. "angel"). Apparently, the name was given, by a person called "Aggelis" who was the owner (gen. Possessor) of this plot. Aggelos > diminutive "Aggelis", a patronymic surname, derived from the baptismal name Aggelos {Αγγελος}. Apparently, from a person called "Aggelis" who was the owner (gen. Possessor) of this plot. |
| 34. | Lyrarakis winery | Ippodromos | Associative | Ippodromos—Ippos (>Eng. "horse") + dromos (>Eng. "road") = Ippodromos (hippodrome). The vineyard at that point is at an elongated flat space, formed with dimensions that match the Roman horse races (that historically existed there). |
| 35. | Lyrarakis winery | Psarades | Descriptive | Psarades—psaros (>Eng. "grey") + ades (>Eng. "the verbal ending for the plural") = psarades (those who have grey color). This area has characteristic grey-colored soils. |
| 36. | Minos Miliarakis Winery | Turtles | Associative | Turtles same place name but different meaning, a settlement located in an area that looks like a turtle shell (the name attributed translated!) |
| 37. | Paraskeva Winery | Lagara | Descriptive | Lagara—lagaros (>Eng. "clear"/"pure"). Metaphorically it means that the element of the plot is pure. |
| 38. | Sclavos Wines | Lacomatia | Associative | Lacomatia— lakos (>Eng. "dip") + matia (<Eng. "eyes= lakomatia = (eyes that gaze into the dip)." The main feature of the whole area is the steep slopes whilst another flat area exists with mild slopes. Due to this orientation, the land gazes at the infinite ocean. |
| 39. | Sclavos Wines | Monampeles | Descriptive | Monampeles—mono (>Eng "only") + ampeli (>Eng "vine") = monampeles (only vines). The only land suited to become a vineyard. |

**Table 1.** *Cont.*

| ID | Winery | Toponymy Label | Toponymy Category | Description Analysis |
|---|---|---|---|---|
| 40. | Sigalas Winery | Kavalieros | Eponymous | Kavalieros—cavalier (>Eng. "master") + eros (Greek suffix indicates an adjective) = Kavalieros. As Santorini was under Frankish rule the locals used to call the previous owner of the vineyard by his authority. There is also another story (probably unfounded): the vineyard owner Mr. George named it "Kavaliero" as it dominates all the other vineyards like a cavalryman. |
| 41. | T-oinos Winery | Stegasta | Descriptive | Stegasta—(> Eng. "stonework low buildings").A type of small roof that covers the topically placed wine press (usually in contact with the vineyards) that workers and residents of Tinos island used in order to protect themselves from the strong winds. Additionally, after the harvest it was practically impossible to transport the grapes through the labyrinthine villages, so they needed to press and ferment the vines on a certain spot of production. A name stemming from the Greek word "στεγαστά" [stéγasta] referring to a kind of stonework low buildings, a type of small roof that covers the topically placed wine press (usually in contact with the vineyards) that workers and residents of Tinos island used in order both to protect themselves from the strong winds and because after the harvest it was practically impossible to transport the grapes through the labyrinthine villages, so they needed to press and ferment the vines on the certain spot of production. |
| 42. | Thymiopoulos Vineyards | Kayafas | Eponymous | Kayafas (Καΐάφας) is Hellenized Hebrew name < Aramaic word Kayphā. Probably lived in this area. |
| 43. | Thymiopoulos Vineyards | Vrana Petra | Descriptive | Vrana Petra—vrana (>Eng. "black") petra (>Eng. "stone") = Vrana Petra (the black stone). One theory considers Vrana from Bulgarian brána (>Eng. "harrow") that entered the Greek language as a vrana, the agricultural wooden/metal tool, used for leveling plowed land. Petra (>Eng. "stone") is a characteristic rock in the area. |
| 44. | Tranampelo Domaine | Tranampelo | Evaluative | Tranampelo—Trano (>Eng. "great") + ampeli (>Eng. "vineyard") = Tranampelo (the great vineyard). The area where vineyards were traditionally cultivated, of special recognition and preference by all the inhabitants of the island. |
| 45. | Tselepos Winery | Kokkinomilos | Descriptive | Kokkinomilos—Kokkino (>Eng. "red") + milos (>Eng. "mill") = Kokkinomilos (the red mill). The vineyard with gravelly red clay soils (loam) that surrounds the watermill gave its characteristic name. |
| 46. | Tselepos Winery | Melissopetra | Descriptive | Melissopetra—Melisso (>Eng. "bee") + petra (>Eng. "rock") = Melissopetra (The bee's rock). The soils are schist (type of rock) and the place used to host beehives. |
| 47. | Tselepos Winery | Avlotopi | Descriptive | Avlotopi—Avlo (>Eng. "reeds") + topi (>Eng. "field") = Avlotopi (the field with many reeds). Reed area, where reeds grow and are suitable for flute construction (flute is a musical instrument mainly referred to by shepherds). The vineyard is close to an affluent River (Dolianitis) where reed grows. |

**Table 1.** *Cont.*

| ID | Winery | Toponymy Label | Toponymy Category | Description Analysis |
|---|---|---|---|---|
| 48. | Tselepos Winery | Marmarias | Descriptive | Marmarias—Marmarias (>Eng. "marble"). The soil in the area up on the hill and after the plot of Kokkimomilos is gravelly clayed with limestone. It also contains significant amounts of marble and the locals gave the characteristic toponymy "Marmarias" as the Greek word {"μάρμαρο"—(marmaro)} stands for marble. |
| 49. | Tselepos Winery | Laoudia | Assosiative | Laoudia—(>Eng. "hares' nests). The name was given by the local dialect of Santorini Island. |

## 7. Findings and Discussion

Condensing the essence of the main conclusions of our research, we have reached the following individual points:

1. The preponderance of wineries use descriptive toponymic names. In Greece, the dominant brands for the Greek wine lover are the wine production zones (i.e., Santorini, Naoussa, etc.) that have been recognized for the quality of the wines produced in them. In our case, the brand (winery) reinforces the primary quality factors such as the place of origin. All wineries that were studied belong to a defined geographical area with either a Protected Designation of Origin (PDO) or Protected Geographical Indication (PGI). We wanted to go one step further, they all recognize the value of a specific vineyard and apply the specific vineyard names on labels (toponymy as a name in their wine label) in order to indicate the exceptional value of their wine stemming from that particular plot since it is the result of careful site and variety matching. They do not translate the toponymy; they opt in favor of using it as it is. There is no doubt that the correct spelling in Modern Greek and its corresponding transliteration into Latin characters is fundamental. The attribution of a place name must be in such a way that it can lead us to "inverted" from the Greek rendering to the true form of the name as closely as possible. It should lead, similarly, to the orthographic expression (and where possible to the pronunciation) of the name. The adoption of ISO 843/ELOT 743 is proper for Greek wine. The Greek state also uses it in the transliteration of addresses {i.e., Greek: "Μεγάλου Αλεξάνδρου" address if formed into Megalou Alexadrou and not Alexander the Great}.

2. The comparative analysis indicated that over half of the toponymy labels are descriptive. We consider this element to be the highest among the studied pricing categories of toponymy wines. Consequently, from the above 49 examined names, 25 (51%) are Descriptive, 15 (31%) are Associative, 6 (12%) are Eponymous, 2 (4%) are Occurrent, 1 (2%) are Evaluative, Shift and Indigenous are zero, Table 2. A significant percentage (51%) of the descriptive criteria indicates what wine producers believe communicates their product to the market. Consumers can become interested in learning about the environmental factors and other contextual characteristics that form and express terroir. Apparently, a descriptive name is more appropriate to transfer the specific terroir to consumers. In this way, a favorable disposition can be developed toward the brand and consequently an actual purchase by consumers. It is interesting though, that there is only one evaluative name in the examined data. This leads to the conclusion that the communicative approach toward the receivers and consumers via emotional allusion is not a preferred method. Yet, we consider that the communication between the wine producer and the possible consumer could be more effective if the name on the wine label focused more on impression, sentiments and feelings. Besides wine is an experiential product and that is exactly the reason why the name rendering by possible consumers is important.

**Table 2.** Toponymy statistical analysis.

| Toponym Type | Statistics | | |
|---|---|---|---|
| | **Frequency (n)** | **Percentage (%)** | **Cumulative Percent** |
| Descriptive | 25 | 51% | 51 |
| Associative | 15 | 31% | 82 |
| Occurrent | 2 | 4% | 86 |
| Evaluative | 1 | 2% | 88 |
| Shift | 0 | 0% | |
| Indigenous | 0 | 0% | |
| Eponymous | 6 | 12% | 100 |
| Total | 49 | 100% | |

3. Place names are divided into two main sections: close compound and non-compound words. It is rather a common tactic, that two words are used together to yield a new meaning and form a new one. For example: in the name Melissopetra, the first synthetic Melisso (>Eng. "bee"), signifies a special feature of the place and the second synthetic petra (>Eng. "rock")}, signifies the place's type. We can conclude at this point and according to our admeasurement, that 33(67%) by 49 names are non-compound words and 16 (33%) names are close compound words. Wine naming is a process based on communication theories that focus on interaction improvement. This could be an indication that noncompound words can contribute better to the communication procedure between the sender (wine producer) and the receiver (consumer).

4. Another element worth mentioning is the analogy between toponymy wine labels and others "regular" labels on each winery to illustrate the divergent paths label naming follows in Table 3.

**Table 3.** Toponymy statistical analysis.

| Winery | Wine Labels Total | Toponymy Labels | Ratio (%) | Details |
|---|---|---|---|---|
| Acroterra Wines | 3 | 1 | 33.33% | |
| Afianes Winery | 13 | 1 | 7.96% | |
| Alpha Estate | 14 | 7 | 50% | The ultra-premium (as winery classifies it) keeps the initial toponymy but the two premiums translate it. Moreover, one toponymy, the translated one (Turtles) is common with one more producer (similar as Turtle) the Minos Winery |
| Artemis Karamolegos Winery | 13 | 2 | 15.38% | |
| Dalamaras Winery | 7 | 1 | 14.29% | This producer share the same place name (Paliokalias) with one more Kokkinos Winery |
| Douloufakis winery | 20 | 2 | 10% | The two labels (one red and one white) share the same place name |
| Hatzidakis Winery | 13 | 2 | 15.38 | |
| Hatzimichalis Domaine | 26 | 6 | 23.08% | |
| Kir-Yianni Estate | 22 | 4 | 18.18 | |
| Kokkinos Winery | 4 | 1 | 25% | This producer shares the same place name (paliokalias) with one more Dalamaras Winery |

**Table 3.** *Cont.*

| Winery | Wine Labels Total | Toponymy Labels | Ratio (%) | Details |
|---|---|---|---|---|
| Lyrarakis winery | 20 | 10 | 50% | |
| Minos Miliarakis Winery | 28 | 2 | 7.14% | |
| Paraskeva Winery | 8 | 1 | 12.5% | |
| Sclavos Wines | 14 | 2 | 14.29% | |
| Sigalas Winery | 10 | 1 | 10% | |
| T-oinos | 8 | 4 | 50% | The same toponymy is shared in the four wine labels (2 red and 2 white) |
| Thymiopoulos Vineyards | 8 | 2 | 25% | |
| Tranampelo Domaine | 1 | 1 | 100% | |
| Tselepos Winery | 17 | 5 | 29.41% | |

Some interesting findings are that we have three (n = 3) wineries having half (50%) of their wine labels named after the place name. The typical ratio for the majority stands between ± 25% for the other wineries. Only one winery uses the same toponymy for the wine label and winery name and also only one has one single label and it is a place name.

We consider the results of this research to be an important basis for redefining the communication strategy of wine companies all over the world, based on the label, and the use of toponymy. Additionally, our intention is to open a discussion about the theoretical models of sense of place and soft power and how wine companies could combine them creating an applicable practical model regarding their communication strategy. However, further study is needed in the future, regarding the practical application of the presented theoretical background, such as researching the consumers' reactions to specific wine labels with specific verbal and non-verbal features.

## 8. Conclusions

Premiumization is a characteristic | future of a product that aggrandizes consumer sensory awareness and subsequently causes the willingness to pay for more. The premiumization of wines has billowed as a buying criterion in the global wine market due to the demand for the highest quality characteristics and value wines. Thinking about premiumization brings to mind higher revenues and greater profitability. The Greek wineries' target group is not purely consumers. They are at the same time the recipients of a communication process, which in order to be successful does not have to be linear but circular. This means that continuous and dynamic communication with the public is imperative and both the theoretical approach of soft power and place branding contribute to this. There are further correlations that can be applied, for example, area designation, price categorization, product characteristics, historical evidence, etc., in order to establish a better understanding of the role of toponymy.

According to the research of the work, we came to the conclusion that the application of soft power and place branding in the communication strategy of Greek wineries can effectively capture the terroir and even more in combination with the toponymy can transform it into a commodified symbol. This means that starting from this base, we could, strengthen the vision for the transformation of wineries into multidimensional cells of cultural, touristic, economic and social activity, with the corresponding results in the wider regions to which they belong.

Wine businesses display particularities, which are good to approach in a more modern and flexible way than the traditional method of action and reaction. The information that can be gleaned from consumer feedback is important to integrate into a more general

understanding of how wine communication will be applied. The toponym label has the potential to be instrumentalized in a modern and competitive environment. This fact can have a long-term effect on the social and economic level of the wider wine region. Their extroversion will be stimulated, as they will be dynamic wine tourism destinations. The important thing is that they will offer an overall touristic experience focused on the history, the terroir, and the special geographical and climatic conditions of each wine region, fully applying the sense of place philosophy. The key to wine communication through toponymy and its connection with wine tourism seems to be directly related to active communication with the public through interaction, engagement, awareness and activation.

Communicating terroir through toponymy seems to be a drastic change, concerning the modern approach of soft power and sense of place, because this approximation has the potential to empower the place branding of the Greek wineries (and not only), and create a new dimension to the terroir through an overall communication framework that will concern the absolute connection of the local identity, the product and the region.

By extension, the consumer—tourist—receiver of communication, can engage with the wine more by creating a relationship of focus, stimulation, empathy and perception of the fact that the label or the wine are symbols of a holistic culture, related to the region and history. At the same time, strategic planning is being established on the part of the wine businesses, oriented towards the creation of a "diverse winery", which offers something much more than a touristic experience: the participation in an active process of two-way communication and conversation with the historicity, sense and culture of a particular place.

**Author Contributions:** Conceptualization, T.T. and E.A.; methodology, T.T., E.A. and G.G.; validation, E.A., V.M.; formal analysis, E.A. and V.M.; investigation, E.A. and V.M.; resources, V.M. and E.A.; writing—original draft preparation, E.A. and T.T.; writing—review and editing E.A., T.T., G.G. and V.M. All authors have read and agreed to the published version of the manuscript.

**Funding:** This research received no external funding.

**Institutional Review Board Statement:** Not applicable.

**Informed Consent Statement:** Not applicable.

**Data Availability Statement:** The data presented in this study are openly available. Additionally, it is possible to contact one of the study authors.

**Conflicts of Interest:** The authors declare no conflict of interest.

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
