# Peer review of "Communicating Terroir through Wine Label Toponymy Greek Wineries Practice"

_sustainability, doi:10.3390/su142316067_

Round 1

Reviewer 1 Report

Dear Author(s),

Thank you very much for this interesting article.

This research seeks to connect soft power and place branding as they reflect properly the communication of toponomy in terms of capturing the terroir of Greek wineries.

According to this, this paper aims to indicate that the application of toponymy as name in wine label helps to tell a wine’s origin story and to transfer the element of terroir.

The abstract needs, in my opinion, a complete rethinking. I couldn’t read, in this section, the main arguments, methodology, aims and the critical contribution of the paper to the field of place branding and soft power.

I recommend to the author(s) to reinforce the literature review section. Especially, and in my opinion, it’s mandatory to create a focus on the term “sense of place”. In my view, this is critical to understand the communication of the Greek’s terroir through Greek’s wine label.

Please take a look at these references:

Bérard, L. (2016). Terroir and the sense of place. In Research handbook on intellectual property and geographical indications. Edward Elgar Publishing.

Kyle, G., & Chick, G. (2007). The social construction of a sense of place. Leisure sciences, 29(3), 209-225.

Massey, D. (2008). A global sense of place. In The cultural geography reader (pp. 269-275). Routledge.

Shamai, S., & Ilatov, Z. (2005). Measuring sense of place: Methodological aspects. Tijdschrift voor economische en sociale geografie, 96(5), 467-476.

This article, in general terms, and according to my view, is too descriptive. It doesn’t exist empirical work. I perfectly understand that is a study case (The terroir of Greek Wineries), but I have serious doubts on the contribution of this article to the field of place branding and soft power.

A critical issue of this article is, in my opinion, the difference between “to comment a case study” or “to analyse a case study”. The current version of the paper offers data and contents that are very close to ‘comment a case study’. For this reason, I firmly recommend some changes to the author(s):

1. Create a discussion and expand the concluding remarks section. It is important to create a valuable debate on the basis of another case studies, and according to the most recent scientific literature that has been published on the relationship between place branding and soft power (I suggested some references on the previous sections of this review).

2. It is necessary to create a strong and clear link between scientific literature and the case study. This is the only way to advance in the knowledge that now exists in the intersection between place branding and soft power. I will be very happy if I have the chance to read more arguments and reflections on this issue in the ‘new’ discussion section and conclusions of this paper.

3. Consider how you present your descriptive data related to the case study analysis, in order to be clearer in differentiating between data and analysis, and to draw out and emphasize your main findings and make sure that these relate to issues and questions identified in the early parts of the paper.

4. This is a very interesting contribution for two different audiences: scholars and practitioners. Please, rewrite your proposal on the basis of both profiles.

For all of these reasons, and in my opinion, the background data of the case is only informative, and also, I think that the case study developed in the paper has an important potential and offers a valuable data. According to this, I really think that the current version of the paper needs to clarify its critical contribution to the relationship between place branding, soft power and sense of place.

I hope that the author(s) finds the energy to apply these suggested major changes to the current version of the paper and do not find my criticism too hard. Good luck.

Author Response

Regarding the revisions of our manuscript, we indicate the following remarks.

Generally,

(I) We checked and confirm that all references are relevant to the contents of the manuscript.
(II) Any revisions to the manuscript are with track changes, in such way that can be easily viewed by the editors and reviewers.
(III) The manuscript undergone an extensive English revisions in order to resolve any issues.

 (IV) Following you will find our reply to certain comments in the review reports

The paper by [Nelson Barber, Joseph Ismail & D. Christopher Taylor (2007) Label Fluency and Consumer Self-Confidence, Journal of Wine Research, 18:2, 73-85] was taken into consideration and added to the manuscript

Reviewer 2 Report

I had the pleasure to review ‘communicating terroir through wine label toponymy: Greek wineries practice’, which I feel has some potential. Authors need to be granted enough time to work anew on this manuscript though in order to efficiently reveal this potential. Main points I’d recommend include the following:

-       - Extremely careful proof-reading, as the vocabulary often employed, grammar, structure of the sentences, paragraphs etc are often incorrect or hard to follow.

-        -The introduction needs to more clearly identify the objective and research gap/ questions.

-        - Refocus the manuscript. As it stands, there is no sufficient justification as to why somebody should explore Greek wineries in specific. A justification can relate particularly to Greek language and the alphabet as the paper focuses on both and neither are easy to follow for a non-Greek speaker (thus the English proverb ‘it’s all Greek to me’). This means restructuring the manuscript and re-organizing information accordingly. An adequate alternative seems to be starting with a section on place brand(ing) and wine branding (here you could decide what your theoretical approach would be (e.g. territorial brand management and sense of place as in Melewar & Skinner (2020), brand heritage as in Kladou, Psimouli & Kapareliotis (2020), something else?), a second section could be on the importance of language and communication (for branding), perhaps followed by a third section (depending on your content and length) focusing on the importance of language and communication for wine branding in specific, where the soft power discourse + place branding aspects come together. Details such as the ones in Table 1 are not that necessary, unless you wish to target a linguistic journal? In any case, these sections together should provide enough support for choosing to focus on Greece as a case. In a methodology section, you start with restating your objective and providing details on your research approach (how you collected data, where, why, when, what type of analysis was necessary – e.g. why these wineries and why these labels), i.e. such as those you refer to at 6. Application of toponymy by Greek wineries. Other information included in your current section 6, however, should move to a ‘findings & discussion’ section. I’d suggest updating and enriching your conclusions section, which should include: (1) conclusions answering each research question/ addressing each research gap clearly, (2) implications for scholars and implications for practitioners (it’d be much better to provide some insights on the general picture and how your work is not relevant to Greek scholars and practitioners only), (3) limitations and future research guidelines.

Author Response

Generally,

(I) We checked and confirm that all references are relevant to the contents of the manuscript.
(II) Any revisions to the manuscript are with track changes, in such way that can be easily viewed by the editors and reviewers.
(III) The manuscript undergone an extensive English revisions in order to resolve any issues.

 (IV) Following you will find our reply to certain comments in the review reports

We checked and reformed:

(1) findings in order to answer each research question

(2) better indication for scholars and implications for practitioners of linking place with wine label in general and not only to Greek market,

(3) updated the limitations and future research guidelines.

Reviewer 3 Report

I would like the authors to take a look at the article, "Label fluency and consumer self-confidence" by Barber, N., et al. and consider consumer self-confidence in their paper. 

Author Response

Generally,

(I) We checked and confirm that all references are relevant to the contents of the manuscript.
(II) Any revisions to the manuscript are with track changes, in such way that can be easily viewed by the editors and reviewers.
(III) The manuscript undergone an extensive English revisions in order to resolve any issues.

 (IV) Following you will find our reply to certain comments in the review reports

The abstract reformed, we tried to emphasise the link between scientific literature and the case study and conclusions of this paper.

The section regarding the “sense of place” is also reformed and the following references suggested by the reviewer was taken into consideration and are included in the manuscript.

Bérard, L. (2016). Terroir and the sense of place. In Research handbook on intellectual property and geographical indications. Edward Elgar Publishing.

Massey, D. (2008). A global sense of place. In The cultural geography reader (pp. 269-275). Routledge.

Shamai, S., & Ilatov, Z. (2005). Measuring sense of place: Methodological aspects. Tijdschrift voor economische en sociale geografie, 96(5), 467-476.

Round 2

Reviewer 1 Report

Dear Authors, Thank you for submitting the revised version of your paper. I'm very happy to announce you that, in my opinion, the paper has been considerably improved.

Author Response

Regarding the revisions of our manuscript, we indicate the following remarks.

Generally,

(I) We checked and confirm that all references are relevant to the contents of the manuscript.
(II) Any revisions to the manuscript are with track changes, in such way that can be easily viewed by the editors and reviewers.
(III) The manuscript undergone an extensive English revision in order to resolve any issues.

 (IV) Following you will find our reply to reviewer suggested minor points to consider:

Concerning the absence of the introduction, it is indeed truly correct and that is why it was added to the new version, as well as the extension of the conclusions, as we have been advised.

Nevertheless, in terms of the potential similarities with the work of Kladou, Psimouli, Kapareliotis (2020), we would kindly recommend that there is no identification in terms of the research field, as the esteemed colleagues have as their main core of interest the family wineries of Crete and examine the possibility of developing branding through the family heritage of each winemaker. In our case, the research is connected to the use of the toponymy as a branding strategy with the ultimate goal of the overall development of each wine region and the increase of wine tourism in it. Also, our effort, in terms of the toponymic method, is based on all Greek wineries, with the perspective of creating, through future research, a broader communicative approach model for every wine company worldwide. This model use the toponymic strategy as well as the theories of sense of place and soft power in terms of the communication of the terroir of each wine zone and will thus be able to transform wineries and regions into cells of economic, commercial, cultural and touristic development.

Regarding the observation about the reduction of the extent of the data presented in sections 1-6, we consider that this extensive material supports efficiently substantial issues for the proper unfolding of our work.  We believe that possible diminution would create difficulty in comprehension of our basic concept and possible alteration of the meaning we want to convey. The enrichment in the specific sections, as it was very correctly pointed out to us in the first comments, is extensive in order to correctly and sufficiently support the theoretical background of the subject we are presenting, that is why we revised it from the very first highlights.

Reviewer 2 Report

Thank you for the revised version. Some minor points to consider:

- Please note that your work seems to be an attempt to address the research gap identified in Kladou, Psimouli, Kapareliotis (2020) on the role of brand architecture for wineries.

- Numbering your discussion in section 7 is not necessary.  

- Why is there no introduction? How about have 1. Introduction and 2. Literature review in which you can have subsections for the theoretical points you raise in current sections 1-5?

- You could shorten parts of the review presented in current sections 1-6 and move this information to section 8 in support of your findings and conclusion. Your conclusion and implications section is too short and superficial and does not connect to the extensive literature review presented earlier in your manuscript.

Author Response

(The authors gave the same response as above.)

Round 3

Reviewer 2 Report

It feels the authors fall in the same fallacy as many winemakers: to (mis)understand the relationship of toponymy with branding (and brand architecture). Had you had the time to more thoroughly read the suggested reading, the contribution of your manuscript would be greater. However, it provides value as it is too - and paves the ground for future studies. Congratulations.